# Nickel Phthalocyanine: Borophene P-N Junction-Based Thermoelectric Generator

**DOI:** 10.3390/ma18122850

**Published:** 2025-06-17

**Authors:** Nevin Taşaltın, İlke Gürol, Cihat Taşaltın, Selcan Karakuş, Bersu Baştuğ Azer, Ahmet Gülsaran, Mustafa Yavuz

**Affiliations:** 1Department of Basic Sciences, Maltepe University, 34857 Istanbul, Türkiye; 2Department of Renewable Energy Technology and Management, Maltepe University, 34857 Istanbul, Türkiye; 3Environment and Energy Technologies Research Center, Maltepe University, 34857 Istanbul, Türkiye; 4Materials Institute, TUBITAK Marmara Research Center, 41470 Gebze, Kocaeli, Türkiye; ilke.gurol@tubitak.gov.tr (İ.G.); cihat.tasaltin@tubitak.gov.tr (C.T.); 5Department of Chemistry, Gebze Technical University, 41400 Gebze, Kocaeli, Türkiye; 6Department of Chemistry, Istanbul University-Cerrahpasa, 34320 Istanbul, Türkiye; selcan@iuc.edu.tr; 7Department of Mechanical and Mechatronics Engineering, University of Waterloo, Waterloo, ON N2L 3G1, Canada; bbastuga@uwaterloo.ca (B.B.A.); agulsaran@uwaterloo.ca (A.G.); myavuz@uwaterloo.ca (M.Y.); 8Waterloo Institute for Nanotechnology (WIN), University of Waterloo, Waterloo, ON N2L 3G1, Canada

**Keywords:** energy, thermoelectric generator, phthalocyanine, nickel phthalocyanine, borophene, nanocomposite

## Abstract

In this study, borophene and nickel phthalocyanine (NiPc): borophene nanocomposites were prepared using the sonication method. The NiPc: borophene nanocomposite was uniformly obtained as a 10–80 nm-sized spherically shaped particle. Electrical conductivities (s) were measured as 3 × 10^−13^ Scm^−1^ and 9.5 × 10^−9^ Scm^−1^ for NiPc and the NiPc: borophene nanocomposite, respectively. The SEM image showed that borophene was homogeneously distributed in the NiPc matrix and increased the charge transport pathways. This is the main reason for a 10^6^-fold increase in electrical conductivity. An indium tin oxide (ITO)/NiPc: borophene nanocomposite-based thermoelectric generator (TEG) was prepared and characterized. The Seebeck coefficients (S) were calculated to be 5 μVK^−1^ and 30 μVK^−1^ for NiPc and the NiPc: borophene nanocomposite, respectively. A positive Seebeck coefficient value for the NiPc: borophene showed the p-type nature of the nanocomposite. The power factors (PF = sS2) were calculated as 7.5 × 10^−16^ μW m^−1^ K^−2^ and 8.6 × 10^−10^ μW m^−1^ K^−2^ for NiPc and the NiPc: borophene nanocomposite, respectively. Compositing NiPc with borophene increased the power factor by ~10^6^-fold. It has been concluded that the electrical conductivity and Seebeck coefficient of the NiPc: borophene material increases due to energy band convergence because of combining p-type NiPc with p-type borophene. Therefore, the NiPc: borophene nanocomposite is a promising material for TEG.

## 1. Introduction

Thermal energy is often an overlooked resource in energy economics. Instead of being a solely consumable by-product of an irreversible labor cycle, it can function as an energy source in its own right. Thermal energy is essential and should not be overshadowed by considerable funding efforts to improve the growth rates of various forms of energy, such as solar energy (photovoltaic), electrical energy storage (batteries and supercapacitors), and chemical energy (low- and high-temperature batteries, electrocatalysis, redox-flow systems, and CO_2_ conversion). Thermoelectric (TE) generators represent solid-state electronic power sources that directly convert thermal energy into electrical energy, providing a crucial resource for future energy economies. Technological advancements in the fields of physics, chemistry, electrical and electronic engineering, and materials science, along with the evolving nature of the energy economy, underscore the crucial role that thermoelectric generators (TEGs) play in harnessing distributed, low-temperature heat sources in the twenty-first century. It is an ideal situation for capturing and transforming energy. In the context of global energy uncertainty and increasing energy demands, improving energy conversion efficiency provides a driving force for energy technologies. TE generators can meet electrical energy requirements as solid-state electronics power sources by converting waste heat into fixed and direct electrical energy using the Seebeck effect [1,2,3,4,5]. Materials must have several specific properties to be thermoelectrically efficient. For efficient thermoelectric (TE) materials, the Seebeck coefficient (S) represents the conversion of a temperature difference in a material into electrical voltage. Mobile charge carriers are distributed along the temperature gradient in the material and play a role in the accumulation of interfacial charge, thus creating a potential difference (ΔV) (Equation (1)). A high S value provides greater electrical energy production. Thermal conductivity (κ) determines a material’s ability to conduct heat. A low κ value increases efficiency by maintaining the thermal gradient. The electrical conductivity (σ) of the material must be high.*S* = Δ*V*/Δ*T*(1)

The figure of merit (ZT) of the material is calculated using (Equation (2))ZT = σS^2^TκZT = κσS^2^T(2)
where *σ* represents the proportionality factor, *κ* represents the thermal conductivity, *T* represents the temperature, V represents generated voltage, *S* represents the Seebeck coefficient, and PF represents the power factor. For an efficient thermoelectric material, ZT > 1 [6,7,8,9].

In energy-efficient technologies, TE materials are selected for their high electrical conductivity, high thermoelectric performance, high Seebeck coefficient, and low thermal conductivity properties. As is known, to obtain a high-efficiency thermoelectric (TE) material, a material with low thermal conductivity, high electrical conductivity, and a high Seebeck coefficient is required; however, it is difficult to find a material that combines these properties. Electrical conductivity is directly correlated with thermal conductivity. Recent evidence suggests that an ideal TE material would have a high Seebeck coefficient and high electrical conductivity [10,11].

In the 1950s, the best thermoelectric (TE) materials were inorganic compounds, such as Bi_2_Te_3_, and the first thermoelectric generators (TEGs) were prepared based on Bi_2_Te_3_ [12,13]. However, the efficiency of the initial Bi_2_Te_3_-based thermoelectric generators (TEGs) was low, leading to the development of TEGs using Bi_2_Te_3_ alloys. Bi_2_Te_3_ alloy-based thermoelectric generators (TEGs) have been reported to operate optimally near room temperature, exhibiting higher efficiency than Bi_2_Te_3_-based TEGs [14]. Despite their improved performance, the use of these materials is limited due to their toxicity. Moreover, the rare earth metal Te increases the cost of the alloys, and its preparation as a TE material requires complex vacuum processes [7,15,16]. Since the 1990s, theoretical calculations on thermoelectric (TE) materials have suggested that TE efficiency can be enhanced by incorporating nanostructures, driving extensive studies in this field [17]. Numerous publications have reported that nanostructures in thermoelectric (TE) materials can improve electrical conductivity and reduce thermal conductivity [18,19,20,21,22,23,24].

Semiconductor polymers exhibit high electrical conductivity, a high Seebeck coefficient, and low thermal conductivity, all of which are essential for good thermoelectric (TE) performance [25,26,27]. Among various TEGs, metal–organic complexes have been extensively developed due to their controllable morphology, processability, long-term operational stability, high reliability, and flexibility [28,29,30,31,32]. Metal–organic polymer-based materials hold great potential for improving the performance of TEGs in waste heat recovery systems due to their high repeatability and carrier mobility [33,34,35,36]. Among them, metal–organic polymers and small molecules are two major categories. While metal–organic polymers have been extensively studied for their TE properties, research on metal–organic small molecules is still in its early stages. However, metal–organic small molecules have several advantages over their polymer counterparts. For instance, they can crystallize more easily while maintaining high carrier mobility, which could be beneficial for TE applications. Due to these advantages, the thermoelectric (TE) properties of metal–organic small molecules are currently being investigated. Metallic phthalocyanines (MPcs), as metal–organic small molecules, are a highly versatile class of organic semiconducting materials renowned for their remarkable chemical and thermal stability. They are composed of a central metal ion surrounded by four isoindole-based ligands, which form a planar macrocycle. MPcs find applications in organic photovoltaics, field-effect transistors, and sensors because they exhibit excellent charge transport and optical properties. Most theoretical studies in the field of metal phthalocyanines have suggested that the Seebeck coefficient of these materials ranges from 0.6 to 1.8 mV/K.

It has been reported in the literature that when a semiconductor organic material is combined with the same type of semiconductor inorganic material, the electrical conductivity and Seebeck coefficient of the newly formed organic–inorganic hybrid material tends to increase [9]. Nickel phthalocyanine (NiPc) is a p-type semiconductor [37,38]. It is a metal–organic small molecule with unique chemical and thermal stability properties [28,39,40,41]. Borophene is a p-type semiconductor, and its electrical conductivity falls within the 10^−6^–10^−7^ S/cm range at room temperature [42,43]. Currently, there is a lack of research on the utilization of borophene as an additive in nanocomposite-based thermoelectric generators (TEGs). The NiPc: borophene combination is an organic–inorganic hybrid material. The aim of this study is to investigate the effect of combining p-type organic NiPc with p-type inorganic borophene on the thermoelectric properties of the NiPc: borophene material, with a focus on energy band convergence.

## 2. Experimental

Boron microparticles (>99.95 purity) were purchased from Nanografi (Ankara, Türkiye). ITO-coated glasses were purchased from Sigma-Aldrich (Taufkirchen, Germany). Borophene was prepared as a p-type material using the sonication method described in our previous publication [44]. NiPc was synthesized following the procedure outlined in our earlier work [45] (Figure 1). Borophene and NiPc were mixed in a 1:1 ratio in chloroform and sonicated in an ultrasound bath for 5 min.

The ITO-coated glass substrate was covered with chemical isolation tape in a patterned manner. Six tapes, each measuring 1 mm in width and 4 cm in length, were glued to the ITO-coated glass, and this sample was placed in a 20:4 HCl:deionized water solution in a horizontal position. After the sample was kept in this solution for 10 h, it was washed with deionized water and dried using nitrogen. When the tapes were removed, the desired ITO pattern for use as an electrode was obtained.

Then, the prepared NiPc: borophene nanocomposite was coated on the patterned ITO-coated glass surface by dropping and drying at room temperature for 2 min. Subsequently, a NiPc: borophene nanocomposite coating was patterned using thin cotton sticks, resulting in six NiPc: borophene nanocomposite strips on the glass. The schematic illustration of the prepared TEG (6-pair n-p material) on a glass substrate is provided in Figure 2. Serial connections were made between the ITO and the NiPc: borophene nanocomposite electrode strips using silver paste. At the end strips on both sides of the TEG, silver paste was also used for the voltage measurement.

Thermoelectric measurements of the prepared thermoelectric generator (TEG) were conducted by suspending it in an air gap between two temperature-controlled blocks—a Peltier module and a block with the exact dimensions. The temperature of the Peltier module (hot side) was adjusted using a heater. The temperature between the hot and cold sides was monitored using Pt100 thermocouples. The thermovoltage between the hot and cold sides of the TEG was measured across the silver paste leads with a GW Instek GDM83-51 (Good Will Instrument Co., Ltd., Taipei, Taiwan). The schema of the electrical circuit of the TEG and the TEG measurement setup are given in Figure 3.

## 3. Results and Discussion

### 3.1. Structural and Chemical Analysis of the NiPc: Borophene Nanocomposite

Borophene was analyzed by X-ray diffraction (XRD) using Cu Kα radiation at 40 kV and 15 mA, with the Highscore Plus XRD program (Version 5.3a) employed. Additionally, high-resolution transmission electron microscopy (HRTEM) and ultraviolet–visible (UV–Vis) diffuse reflectance spectroscopy (using a Shimadzu UV-2501PC spectrometer, Shimadzu Corporation (Kyoto, Japan)) were employed (Figure 4). HRTEM images with low magnification are presented in Figure 4a and those with high magnification in Figure 4b, both in DMF medium. The photos indicate that the obtained borophene is in the form of nanosheets, exhibiting a crystal structure with a d-spacing of 0.41 nm. This aligns with the characteristics of a β-rhombohedral boron structure. Furthermore, an individual borophene nanolayer was examined using fast Fourier transform (FFT), as shown in the inset of Figure 4b. The XRD peaks, analyzed in Figure 4c, are consistent with the (0001) plane of a β-rhombohedral boron (PDF31-0207). UV–Vis diffuse reflectance spectroscopy was utilized to analyze band gap energies, with the Kubelka–Munk function employed to process the obtained data using the formula(3)$$F\left(R\right)=\frac{{\left(1-R\right)}^{2}}{2R}$$
where R represents the diffuse reflectance value derived from the measurements. The Tauc plot of the Kubelka–Munk function was then applied, revealing both direct and indirect band gap transitions of the borophene sample, as illustrated in Figure 4d. The calculated band gap energies for the synthesized borophene are 0.94 eV and 1.74 eV for indirect and direct transitions, respectively.

The NiPc: borophene nanocomposite was characterized by scanning electron microscopy (SEM) (FEI Quanta 450 model, Hillsboro, Oregon, USA, conditions: a 6–10 mm working distance, 0–130 Pa pressure, and voltage of 7–10 kV under a low-vacuum medium) and Fourier transform infrared (FTIR) spectroscopy (Perkin Elmer, Waltham, MA, USA, Spectrum Two model, in the 4000–400 cm^−1^ frequency range) (Figure 5). As shown in Figure 5a,b, the SEM image reveals that borophene is homogeneously distributed in the NiPc matrix, thereby increasing the charge transport pathways. This is the main reason for the 10^6^-fold increase in electrical conductivity. The NiPc: borophene nanocomposite was uniformly obtained as a 10–80 nm spherical-shaped particle due to the borophene nanosheet arrangement with the NiPc. In Figure 5c, the FTIR spectra of the borophene, NiPc, and NiPc: borophene are shown, highlighting the functional groups of all samples. In our previous study, the characteristic peaks of borophene were reported. These results showed that the distinctive peaks of borophene were observed at 3479 cm^−1^ (O-H), 2929 cm^−1^ (B-B), 2861 cm^−1^ (B-H), 1653 cm^−1^ (C=O), 1496 cm^−1^ (B-H), 1385 cm^−1^ (B-O), 1255 cm^−1^ (B-O), and 1091 cm^−1^ (B-O-B vibrations). The peaks of the prepared NiPc were observed at 3504 cm^−1^ (N–H stretching), 2920 cm^−1^ (C–H stretching), and 1680 cm^−1^ (C=N/C=C stretching). Additionally, the characteristic peaks of the prepared NiPc were observed at 1090 cm^−1^, 845 cm^−1^, and 756 cm^−1^ and are attributed to phthalocyanine skeletal vibrations. The characteristic peaks of the prepared NiPc: borophene were found at 2922 cm^−1^ (C–H), 1462 cm^−1^ (–CN group), and 748 cm^−1^ (bending modes of vibration of nickel) [45]. Consequently, the appearance of the –CN group at 1462 cm^−1^ and the disappearance of the –NH peak at 3150 cm^−1^ in the structure confirmed the formation of NiPc: borophene.

### 3.2. Thermoelectric Performance of the Prepared TEGs

Measurements of electrical conductivity and the Seebeck coefficient for NiPc and NiPc: borophene were taken (Figure 6). Electrical conductivities (s) were measured to be 3 × 10^−13^ Scm^−1^ and 9.5 × 10^−9^ Scm^−1^ for NiPc and the NiPc: borophene nanocomposite, respectively (Figure 6a). The Seebeck coefficients (S) of the NiPc and the NiPc: borophene nanocomposite were measured as 5 mVK^−1^ and 30 mVK^−1^, respectively (Figure 6b). Borophene can enhance charge transport in the NiPc matrix without disrupting the crystal structure. NiPc is a planar, π-conjugated organic semiconductor. When composited with borophene, efficient charge injection can occur at the interfaces. NiPc molecules stack in columnar structures with π–π interactions along the stacking axis (a/b plane). Because borophene is a two-dimensional structure, it can align parallel to the π system of NiPc, facilitating lateral charge delocalization without disrupting the molecular packing. Because borophene is a p-type semiconductor, it can increase hole mobility by attracting charge from the Ni atoms of NiPc [46]. Borophene increased the electrical conductivity of the hybrid material while only causing a slight increase in the thermal conductivity of the NiPc: borophene hybrid material.

These results reveal that both NiPc and the NiPc: borophene nanocomposite were p-type semiconductors. The power factors (PF = σS^2^) were calculated as 7.5 × 10^−16^ μW m^−1^ K^−2^ and 8.6 × 10^−10^ μW m^−1^ K^−2^ for NiPc and the NiPc: borophene nanocomposite, respectively (Figure 6c). Compositing NiPc with borophene increased the power factor by ~10^6^-fold. It has been concluded that the electrical conductivity and Seebeck coefficient of the NiPc: borophene material increases due to energy band convergence due to combining p-type NiPc with p-type borophene. A comparison of the power factor of the prepared TEGs with that of previous studies is presented in Table 1. This study is the first to investigate the thermoelectric properties of NiPc: borophene. It provides a basis for demonstrating the potential of borophene to increase the thermoelectric performance of organic or polymer materials and contributes to the literature. Although the obtained PF values fall short of those of some high-performance materials, such as those listed in Table 1, it may be possible to increase the PF values in the future through electrical conductivity or structural optimizations.

## 4. Conclusions

In this study, borophene and nickel phthalocyanine (NiPc)-borophene nanocomposites were prepared using the sonication method. The NiPc: borophene nanocomposite was uniformly obtained as 10–80 nm-sized spherical particles. An ITO/NiPc: borophene nanocomposite-based TEG was prepared and characterized. These results indicate that both NiPc and NiPc-borophene nanocomposites exhibit p-type semiconductor properties. The power factors (PF = σS^2^) were calculated as 7.5 × 10^−8^ mW m^−1^ K^−2^ and 8.6 × 10^−2^ mW m^−1^ K^−2^ for NiPc and the NiPc: borophene nanocomposite, respectively. Because borophene enhances the electrical conductivity of NiPc, the NiPc: borophene nanocomposite exhibits a higher power factor. This increase in conductivity suggests that borophene doping would lead to a significant improvement in TE efficiency due to its high electrical conductivity. The production of thermoelectric materials at a low cost and through a simple process that can be achieved from a solution is expected to reduce energy costs and enhance welfare, offering a solution to significant challenges in sustainable energy production.

## Figures and Tables

**Figure 1 materials-18-02850-f001:**
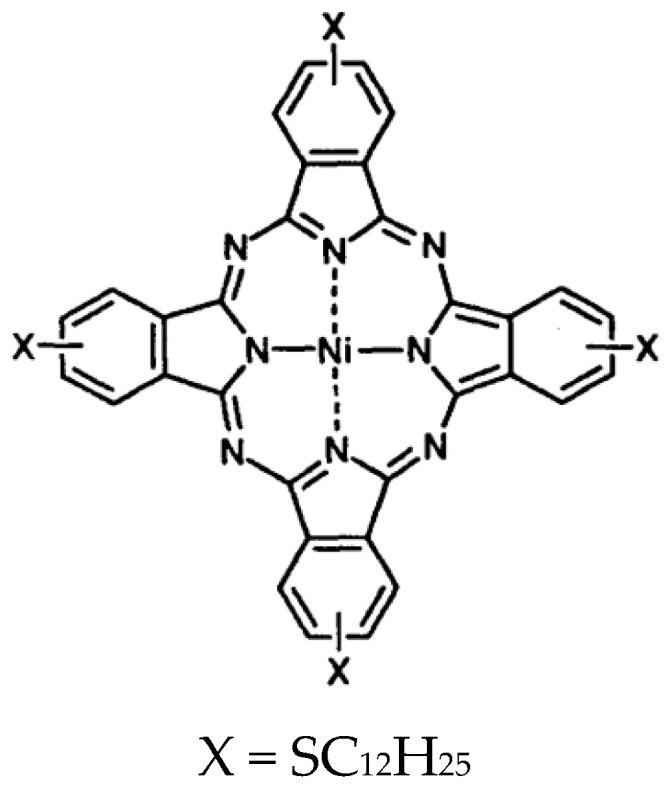
Structure of the synthesized NiPc.

**Figure 2 materials-18-02850-f002:**
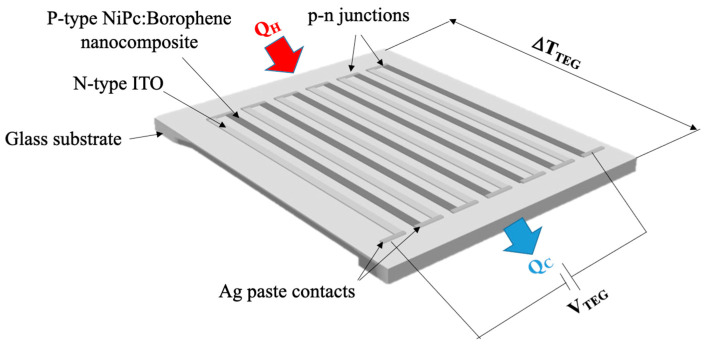
Schematic illustration of the prepared TEG.

**Figure 3 materials-18-02850-f003:**
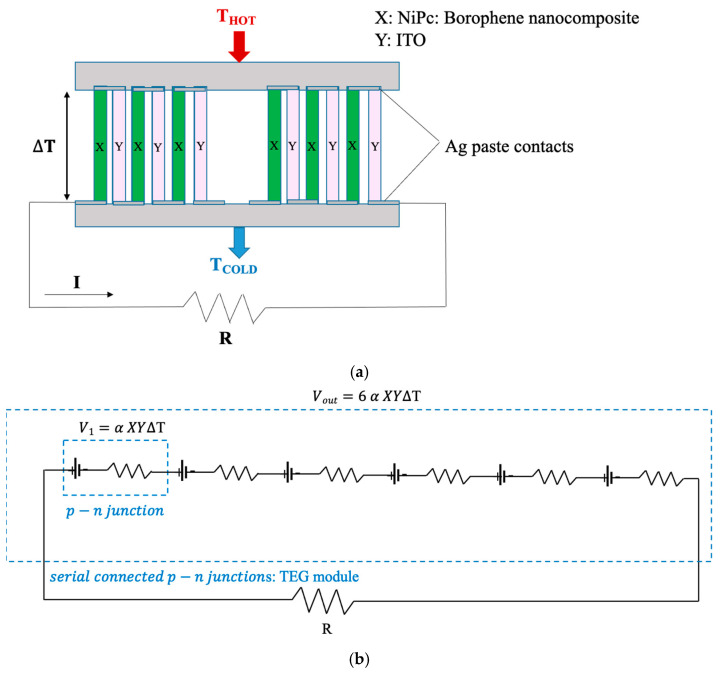
Schematic illustration of (**a**) the electrical circuit of the TEG, and (**b**) the TEG measurement setup.

**Figure 4 materials-18-02850-f004:**
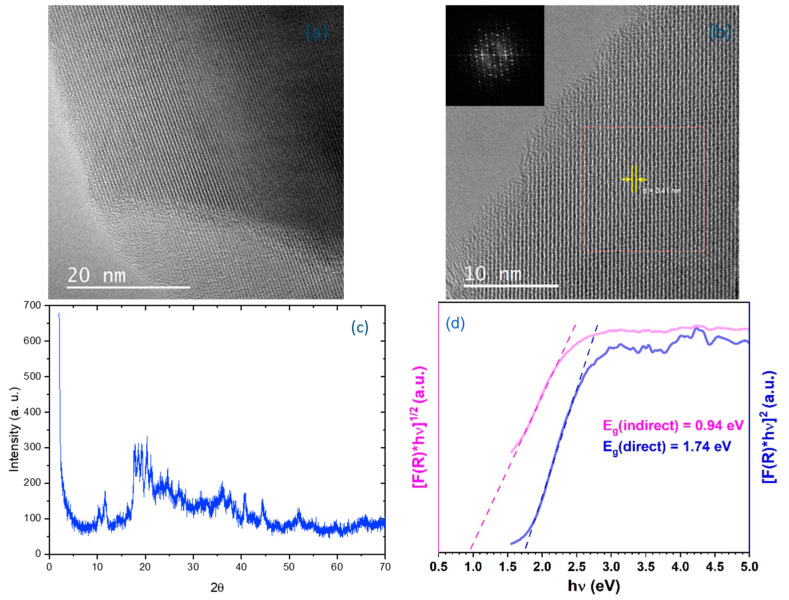
(**a**) Low-magnification, (**b**) high-magnification HRTEM images of the freestanding borophene nanosheets, (**c**) XRD analysis of the borophene nanosheets, and (**d**) the Tauc plots of the Kubelka–Munk function display both direct and indirect bandgaps of the prepared borophene sample.

**Figure 5 materials-18-02850-f005:**
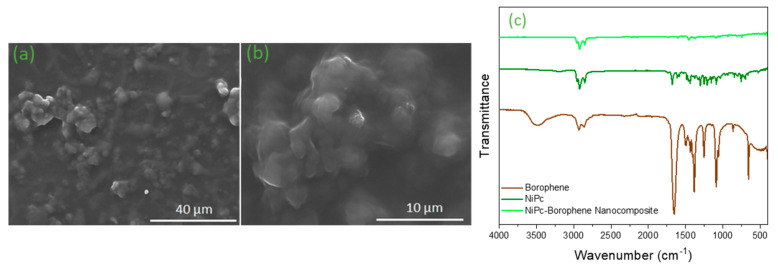
SEM images of the prepared (**a**) NiPc: borophene (1300×) and (**b**) NiPc: borophene (5700×), FTIR spectras of (**c**) borophene, NiPc, and NiPc: borophene.

**Figure 6 materials-18-02850-f006:**
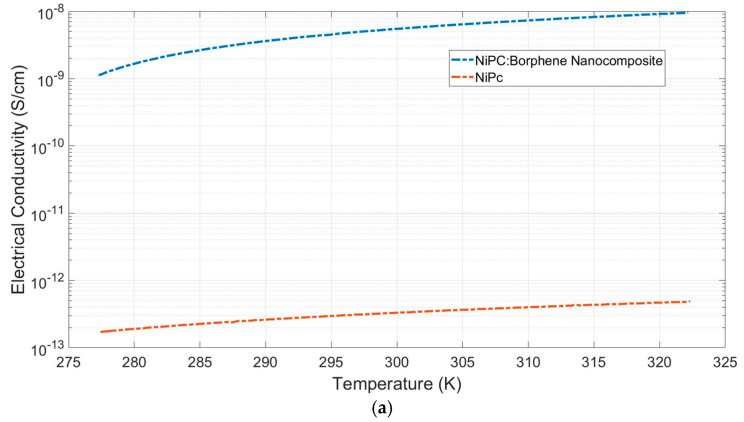
(**a**) Electrical conductivity, (**b**) Seebeck coefficient, and (**c**) power factor versus temperature of NiPc and the NiPc: borophene nanocomposite.

**Table 1 materials-18-02850-t001:** Comparison of the power factor of the prepared TEGs with previous studies.

Compound	S (μV K^−1^)	σ (Scm^−1^)	PF (μW m^−1^ K^−2^)	Ref.
ZnPc	285	8 × 10^−7^	6 × 10^−6^	[47]
CoPc(AsF_6_)_0.5_	50	1 × 10^2^	25	[48]
NiPc(AsF_6_)_0.5_	25	1 × 10^3^	62	[49]
NiPc/SWCNT with 80 wt% SWCNTs	48.5	540	120	[50]
CuPc/SWCNT with 80 wt% SWCNTs	47	450	90	[50]
CoPc/SWCNT with 80 wt% SWCNTs	46	380	80	[50]
NiPc NiPc: borophene	5 30	3 × 10^−13^ 9.5 × 10^−9^	7.5 × 10^−16^ 8.6 × 10^−10^	This study

## Data Availability

The original contributions presented in this study are included in the article. Further inquiries can be directed to the corresponding author.

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
