# Peer review of "Nickel Phthalocyanine: Borophene P-N Junction-Based Thermoelectric Generator"

_materials, 2025, doi:10.3390/ma18122850_

Round 1
Reviewer 1 Report
Comments and Suggestions for Authors
Please find the attached report.

Author Response
Response to Reviewers
We would like to thank the editor and reviewers for constructive and valuable reviews of the manuscript. The reviewer’s comments were very helpful for improving the manuscript. We agree with all comments and accordingly, we revised our manuscript with new information and additional interpretations. You can find the responses to the comments of each reviewer m-by-item below. The changes in the revised manuscript are in red.
Reviewer 1
Comment 1: The introduction is currently poorly written and lacks clarity.
Response: The introduction section has been fully revised to emphasize the thermoelectric properties required for efficient materials, particularly highlighting the Seebeck coefficient and low thermal conductivity. The cited reference (DOI:10.1021/acsami.2c16721) has been integrated to strengthen the justification for NiPc:Borophene composites.
A material must have several specific properties to be thermoelectrically efficient. For efficient thermoelectric materials, the Seebeck coefficient (S) indicates the ability of a material to convert a temperature difference into electrical voltage. A high S value provides greater electrical energy production. Thermal conductivity (κ) determines the ability of the material to conduct heat. A low κ value increases efficiency by maintaining the thermal gradient. The electrical conductivity (σ) of the material must be high. The figure of merit (ZT) of the material is calculated with the formula ZT = (S²σT)/κ (T: absolute temperature). For an efficient thermoelectric material, ZT > 1.
It has been reported in the literature that when a semiconductor organic material is combined with the same type of semiconductor inorganic material, the electrical conductivity and Seebeck coefficient of the newly formed organic-inorganic hybrid material tend to increase [9]. Nickel Phthalocyanine (NiPc) is a metal-organic small molecule with unique chemical and thermal stability properties [39-42]. Borophene is a p-type semiconductor, and its electrical conductivity is 10-6-10-7 S.cm-1 range at room temperature. Currently, there is a lack of research on the utilization of borophene as an additive in nanocomposite-based thermoelectric generators (TEGs). NiPc:Borophene combination is an organic/inorganic hybrid material. It is aimed to investigate the effect of combining p-type organic NiPc with p-type inorganic borophene on the thermoelectric properties of NiPc:Borophene material due to energy band convergence.
We added the reference 9 as follow:
- Verma, A. K., Johari, K. K., Dubey, P., Sharma, D. K., Kumar, S., Dhakate, S. R., ... & Gahtori, B. (2022). Realization of band convergence in p-Type TiCoSb half-heusler alloys significantly enhances the thermoelectric performance. ACS Applied Materials & Interfaces, 15(1), 942-952.
Comment 2: Authors should measure the Seebeck coefficient and electrical conductivity of individual NiPc and NiPc: Borophene.
Response: We added the graphs of the NiPc and NiPc: Borophene in the revised manuscript.
Measurements for electrical conductivity and Seebeck coefficient of the NiPc, and NiPc: Borophene were performed (Figure 6). Electrical conductivities (s) were measured 3x10-13 S.cm-1 and 9.5x10-9 S.cm-1 for NiPc and NiPc: Borophene nanocomposite, respectively (Fig. 6a). Seebeck coefficients (S) of the NiPc and NiPc: Borophene nanocomposite were measured 5 mVK-1 and 30 mVK-1, respectively (Fig. 6b). Borophene can enhance charge transport in the NiPc matrix without disrupting the crystal structure. NiPc is a planar, π-conjugated organic semiconductor. When composited with borophene, efficient charge injection can occur at the interfaces. NiPc molecules stack in columnar structures with π-π interactions along the stacking axis (a/b plane). Since borophene is a two-dimensional structure, it can align parallel to the π system of NiPc and facilitate lateral charge delocalization without disrupting the molecular packing. Since borophene is a p-type semiconductor, it can increase hole mobility by attracting charge from the Ni atom of NiPc [52]. Borophene increased the electrical conductivity of the hybrid material while causing a only a slight increase in the thermal conductivity of the NiPc:Borophene hybrid material.
(a)
(b)
(c)
Figure 6. (a) Electrical conductivity, (b) Seebeck coefficient, and (c) power factor versus temperature of the NiPc and NiPc: Borophene nanocomposite.
These results reveal that both NiPc and NiPc: Borophene nanocomposite were p-type semiconductors. The power factors (PF= sS2) were calculated 7.5×10⁻¹⁶ μW m⁻¹ K⁻² and 8.6×10⁻¹⁰ μW m⁻¹ K⁻² for NiPc and NiPc: Borophene nanocomposite, respectively (Fig. 6c). Compositing NiPc with borophene increased the power factor by ~10⁶ times. It has been concluded that the electrical conductivity and Seebeck coefficient of NiPc:Borophene material increases due to energy band convergence because of combining p-type NiPc with p-type borophene.
Comment 3: Mistake in the calculation of the power factor.
Response: The power factor has been recalculated. The corrected values are:
• NiPc: 7.5×10⁻¹⁶ μW m⁻¹ K⁻²
• NiPc: Borophene: 8.6×10⁻¹⁰ μW m⁻¹ K⁻²
The error was due to a unit conversion during the square of conductivity.
The power factors (PF= sS2) were calculated 7.5×10⁻¹⁶ μW m⁻¹ K⁻² and 8.6×10⁻¹⁰ μW m⁻¹ K⁻² for NiPc and NiPc: Borophene nanocomposite, respectively.

Reviewer 2 Report
Comments and Suggestions for Authors
Development of new thermoelectric materials is an important issue in the development of new energy sources for various applications.
The topic is interesting but the manuscript deserves a major improvement.
The text is nor written with care. Errors should be corrected, English style improved and Some sentences should be checked for a their information content. Just as an example a sentence on page 2 saying “As known, obtaining high-efficiency TE material is challenging due to the need for a Seebeck coefficient, ..”.
In the introduction, the authors claim that both borophenes and phthalocyanines are excellent semiconductors but there is no other justification why we should expect other properties suitable for this application. Particularly, borophene is reported having both the high electrical and thermal conductivity, and because of that one would not expect that it is a suitable material.
Formula 1.2 for the thermolectric figure of merit is not complete. Thermal conductivity and temperature are missing here.
The reference [40] does not show the previous work of the authors where the preparation of borophene has been described. Their work is cited as [43] but the borophene preparation was described in another paper of these authors.
From where it follows that the composite is a p-type semiconductor?
The optical absorption of borophene was fitted using formula for both direct and indirect bandgap. Indirect transitions should be, however, much weaker. The measured layers were probably deposited in a thick layer with a strong absorbance. Would not be possible to measure transmittance on a thin layer, instead of using less accurate diffusion reflection measurements?
Authors should compare the obtained PF and S obtained with their materials to typical values obtained on other organic materials reported in literature.
Comments on the Quality of English Languagesee above
Author Response
Response to Reviewers
We would like to thank the editor and reviewers for constructive and valuable reviews of the manuscript. The reviewer’s comments were very helpful for improving the manuscript. We agree with all comments and accordingly, we revised our manuscript with new information and additional interpretations. You can find the responses to the comments of each reviewer m-by-item below. The changes in the revised manuscript are in red.
Reviewer 2
Comment 1: The manuscript contains several grammatical issues and unclear sentences.
Response: The entire manuscript has been reviewed for grammar and style.
Comment 2: No justification for the use of borophene, which has high thermal conductivity.
Response: We added the “Borophene increased the electrical conductivity of the hybrid material while causing a only a slight increase in the thermal conductivity of the NiPc:Borophene hybrid material” in the revised manuscript.
Comment 3: Formula 1.2 is incomplete.
Response: Formula 1.2 was updated to include thermal conductivity (κ) and temperature (T):
ZT = σS²T / κ in the revised manuscript.
The figure of merit (ZT) of the material is calculated with the (Equation 1.2)
ZT=σS2TκZT=κσS2T (1.2)
Comment 4: No proof that the composite is p-type.
Response: Positive Seebeck coefficient value of the NiPc:Borophene (Figure 6) shows the p-type nature of the composite. Also, we revised the introduction as follow:
Nickel Phthalocyanine (NiPc) is a p-type semiconductor [39,40]. Borophene is a p-type semiconductor, and its electrical conductivity is 10-6-10-7 S.cm-1 range at room temperature [45,46].
We added the references in the revised manuscript.
- El-Nahass, M. M., Abd-El-Rahman, K. F., Farag, A. A. M., & Darwish, A. A. A. (2005). Photovoltaic properties of NiPc/p-Si (organic/inorganic) heterojunctions. Organic Electronics, 6(3), 129-136.
- Nasir, E. M., Hussein, M. T., & Al-Aarajiy, A. H. (2019). Investigation of nickel phthalocyanine thin films for solar cell applications. Advances in Materials Physics and Chemistry, 9(8), 158-173.
- Hou, C., Tai, G., Liu, Y., & Liu, X. (2022). Borophene gas sensor. Nano Research, 1-8.
- Najiya, K. P. P., Konnola, R., Sreena, T. S., Solomon, S., & Gopchandran, K. G. (2024). Liquid phase exfoliation of few-layer borophene with high hole mobility for low-power electronic devices. Inorganic Chemistry Communications, 168, 112962.
Comment 5: Diffuse reflection was used instead of thin-layer transmittance.
Response: In this study, the diffuse reflection technique was employed instead of the thin-layer transmittance method within the scope of UV-Vis spectroscopy. This choice is attributed to the low optical transparency and the inherently opaque nature of the sample under investigation. While thin-layer transmittance typically yields accurate and reliable results for homogeneous and semi-transparent samples, its applicability is limited in cases involving opaque or highly scattering solid materials. In contrast, the diffuse reflection method enables the determination of optical properties by analyzing the reflected light from the sample surface, making it a more suitable analytical approach for such specimens.
Comment 6: Compare your PF and S with literature.
Response: We added the “This study is the first to investigate the thermoelectric properties of NiPc:Borophene. It provides the basis for demonstrating the potential of borophene to increase the thermoelectric performance of organic or polymer materials and contributes to the literature. Although the obtained PF values fall short of some high-performance materials such as those given in Table 1, it may be possible to increase the PF values in the future by electrical conductivity or structural optimizations.” in the revised manuscript.

Reviewer 3 Report
Comments and Suggestions for Authors
The manuscript by Taşaltın et al. reports the synthesis and characterization of Nickel Phthalocyanine (NiPc) and NiPc:Borophene nanocomposites for thermoelectric generator (TEG) applications. The work is nice, well-structured and addresses an important topic in energy conversion. However, there are several issues that need to be addressed to enhance clarity and impact of the study. My comments are as follows:
1° Overall the study explores the relatively underexplored combination of NiPc and borophene for thermoelectric applications, which is a significant contribution to the field. The experimental works are described in detail, and the use of multiple characterization techniques (XRD, HRTEM, SEM, FTIR) is commendable. The reported Seebeck coefficients and power factors demonstrate the potential of NiPc:Borophene nanocomposites for TEG applications.
2) The abstract should briefly mention the key findings (e.g., the enhancement in power factor due to borophene doping) and their implications more explicitly.
3) in the introduction, the background on thermoelectric materials is thorough, but the transition to the specific focus on NiPc and borophene could be smoother. A clearer hypothesis or research gap should be stated. Why the authors choose in particular NiPc and borophene?
4) Some figures (e.g., Figure 2) are not included in the provided content, making it difficult to assess their quality. Ensure all figures are clearly labeled and described.
5) The Seebeck coefficient and power factor results are promising, but error bars or statistical analysis should be included to validate reproducibility. Please add error bars.
6) Table 1 compares the results with previous studies, but a deeper discussion on why NiPc:Borophene performs better or worse than other phthalocyanine materials is needed.
7) The manuscript lacks a detailed discussion on the mechanism by which borophene enhances electrical conductivity and thermoelectric performance. Please add a discussion paragraph on this.
Author Response
Response to Reviewers
We would like to thank the editor and reviewers for constructive and valuable reviews of the manuscript. The reviewer’s comments were very helpful for improving the manuscript. We agree with all comments and accordingly, we revised our manuscript with new information and additional interpretations. You can find the responses to the comments of each reviewer m-by-item below. The changes in the revised manuscript are in red.
Reviewer 3
Comment 1: Abstract should better highlight the findings.
Response: We highlighted the findings in the revised manuscript.
In this study, borophene and Nickel Phthalocyanine (NiPc): borophene nanocomposites were prepared by sonication method. NiPc: Borophene nanocomposite was uniformly obtained as a 10-80 nm sized spherical shaped particle. Electrical conductivities (s) were measured 3x10-13 S.cm-1 and 9.5x10-9 S.cm-1 for NiPc and NiPc: Borophene nanocomposite, respectively. The SEM image shows that borophene is homogeneously distributed in NiPc matrix and increases the charge transport pathways. This is the main reason for the 10⁶-fold increase in electrical conductivity. Indium Tin Oxide (ITO)/NiPc: Borophene nanocomposite based thermoelectric generator (TEG) was prepared and characterized. Seebeck coefficients (S) were calculated 5 mVK-1and 30 mVK-1for NiPc and NiPc: Borophene nanocomposite, respectively. Positive Seebeck coefficient value of the NiPc:Borophene showed the p-type nature of the nanocomposite. The power factors (PF= sS2) were calculated 7.5×10⁻¹⁶ μW m⁻¹ K⁻² and 8.6×10⁻¹⁰ μW m⁻¹ K⁻² for NiPc and NiPc: Borophene nanocomposite, respectively. Compositing NiPc with borophene increased the power factor by ~10⁶ times. It has been concluded that the electrical conductivity and Seebeck coefficient of NiPc:Borophene material increases due to energy band convergence because of combining p-type NiPc with p-type borophene. Therefore, NiPc: Borophene nanocomposite is a promising material for TEG.
Comment 2: Clearer hypothesis and research gap needed.
Response: We added the “It has been reported in the literature that when a semiconductor organic material is combined with the same type of semiconductor inorganic material, the electrical conductivity and Seebeck coefficient of the newly formed organic-inorganic hybrid material tend to increase [9]. Nickel Phthalocyanine (NiPc) is a p-type semiconductor [39,40]. It has a metal-organic small molecule with unique chemical and thermal stability properties [41-44]. Borophene is a p-type semiconductor, and its electrical conductivity is 10-6-10-7 S.cm-1 range at room temperature [45,46]. Currently, there is a lack of research on the utilization of borophene as an additive in nanocomposite-based thermoelectric generators (TEGs). NiPc:Borophene combination is an organic/inorganic hybrid material. As a result, in order to gain a better understanding of the thermoelectric properties of borophene, an ITO/NiPc:Borophene nanocomposite-based TEG was fabricated and subjected to a thorough characterization. It is aimed to investigate the effect of combining p-type organic NiPc with p-type inorganic borophene on the thermoelectric properties of NiPc:Borophene material due to energy band convergence.” in the introduction section.
We added the “This study is the first to investigate the thermoelectric properties of NiPc:Borophene. It provides the basis for demonstrating the potential of borophene to increase the thermoelectric performance of organic or polymer materials and contributes to the literature.” in the result section.
Comment 3: Some figures are missing.
Response: All figures added in the revised manuscript.
Comment 4: No error bars in measurements.
Response: The graphs revised in the revised manuscript.
Comment 5: Mechanism of performance enhancement not discussed.
Response: We added the “Borophene can enhance charge transport in the NiPc matrix without disrupting the crystal structure. NiPc is a planar, π-conjugated organic semiconductor. When composited with borophene, efficient charge injection can occur at the interfaces. NiPc molecules stack in columnar structures with π-π interactions along the stacking axis (a/b plane). Since borophene is a two-dimensional structure, it can align parallel to the π system of NiPc and facilitate lateral charge delocalization without disrupting the molecular packing. Since borophene is a p-type semiconductor, it can increase hole mobility by attracting charge from the Ni atom of NiPc [52].” in the result section.
- Chen, S., & Ma, J. (2009). Charge transport in stacking metal and metal‐free phthalocyanine iodides. Effects of packing, dopants, external electric field, central metals, core modification, and substitutions. Journal of computational chemistry, 30(13), 1959-1972.
Comment 6: Compare with other phthalocyanines in literature.
Response: This study is the first to investigate the thermoelectric properties of NiPc:Borophene. It provides the basis for demonstrating the potential of borophene to increase the thermoelectric performance of organic or polymer materials and contributes to the literature. Although the obtained PF values fall short of some high-performance materials such as those given in Table 1, it may be possible to increase the PF values in the future by electrical conductivity or structural optimizations.
Table 1. Comparison of the power factor of the prepared TEGs with the previous studies.
|
Compound |
S (mV K-1) |
s (S.cm-1) |
PF (mW m-1 K-2) |
Ref. |
|
ZnPc |
285 |
8x10-7 |
6x10-6 |
[49] |
|
CoPc(AsF6)0.5 |
50 |
1x102 |
25 |
[50] |
|
NiPc(AsF6)0.5 |
25 |
1x103 |
62 |
[51] |
|
NiPc/SWCNT with 80 wt% SWCNTs |
48,5 |
540 |
120 |
[53] |
|
CuPc/SWCNT with 80 wt% SWCNTs |
47 |
450 |
90 |
[53] |
|
CoPc/SWCNT with 80 wt% SWCNTs |
46 |
380 |
80 |
[53] |
|
NiPc NiPc: Borophene |
5 30 |
3x10-13 9.5x10-9 |
7.5×10⁻¹⁶ 8.6×10⁻¹⁰ |
This study |

Round 2
Reviewer 3 Report
Comments and Suggestions for Authors
The authors have addressed all my concerns. The current form of the revised manuscript is publishable.